# Urgent need for consistent standards in functional enrichment analysis

**Kaumadi Wijesooriya**[1], **Sameer A. Jadaan**[2], **Kaushalya L. Perera**[1], **Tanuveer Kaur**[1], **Mark Ziemann**[1]*

**1** Deakin University, School of Life and Environmental Sciences, Geelong, Australia, **2** College of Health and Medical Technology, Middle Technical University, Baghdad, Iraq

* m.ziemann@deakin.edu.au

## Abstract

Gene set enrichment tests (a.k.a. functional enrichment analysis) are among the most frequently used methods in computational biology. Despite this popularity, there are concerns that these methods are being applied incorrectly and the results of some peer-reviewed publications are unreliable. These problems include the use of inappropriate background gene lists, lack of false discovery rate correction and lack of methodological detail. To ascertain the frequency of these issues in the literature, we performed a screen of 186 open-access research articles describing functional enrichment results. We find that 95% of analyses using over-representation tests did not implement an appropriate background gene list or did not describe this in the methods. Failure to perform p-value correction for multiple tests was identified in 43% of analyses. Many studies lacked detail in the methods section about the tools and gene sets used. An extension of this survey showed that these problems are not associated with journal or article level bibliometrics. Using seven independent RNA-seq datasets, we show misuse of enrichment tools alters results substantially. In conclusion, most published functional enrichment studies suffered from one or more major flaws, highlighting the need for stronger standards for enrichment analysis.

## Author summary

Functional enrichment analysis is a commonly used technique to identify trends in large scale biological datasets. In biomedicine, functional enrichment analysis of gene expression data is frequently applied to identify disease and drug mechanisms. While enrichment tests were once primarily conducted with complicated computer scripts, web-based tools are becoming more widely used. Users can paste a list of genes into a website and receive enrichment results in a matter of seconds. Despite the popularity of these tools, there are concerns that statistical problems and incomplete reporting are compromising research quality. In this article, we conducted a systematic examination of published enrichment analyses and assessed whether (i) any statistical flaws were present and (ii) sufficient methodological detail is provided such that the study could be replicated. We found that lack of methodological detail and errors in statistical analysis were widespread, which undermines the reliability and reproducibility of these research articles. A set of best practices is urgently needed to raise the quality of published work.

**Data Availability Statement:** All data and code to suport this work is available from GitHub (https://github.com/markziemann/SurveyEnrichmentMethods). We have also used

Zenodo to assign a DOI to the repository: 10.5281/zenodo.5763096.

**Funding:** The authors received no specific funding for this work.

**Competing interests:** The authors have declared that no competing interests exist.

## Introduction

Since the turn of the millennium, high throughput "omics" techniques like microarrays and high throughput sequencing have brought with them a deluge of data. These experiments involve the measurement of thousands of genes simultaneously and can identify hundreds or even thousands of significant associations in a single experiment. Interpreting such data is extraordinarily challenging, as the sheer number of associations can be difficult to investigate in a gene-by-gene manner. Instead, many tools have been developed to summarize regulated gene expression profiles into simplified functional categories. These functional categories typically represent signaling or biochemical pathways, curated from information present in the literature, hence the name functional enrichment. The validity of functional enrichment analysis is dependent upon rigorous statistical methods as well as accurate and up-to-date gene functional annotations.

Two of the most frequently used databases of gene annotations are Gene Ontology (GO) and Kyoto Encyclopedia of Genes and Genomes (KEGG). Both databases emerged around the time of public release of the first eukaryotic genomes, with the aim of systematically cataloging gene and protein function [1,2].

Widely used functional enrichment tools can be classified into two main categories; (i) over-representation analysis (ORA) and (ii) functional class scoring (FCS), and the most common application is in differential gene expression analysis. In ORA, differentially expressed genes (DEGs) meeting a significance and/or fold change threshold are queried against curated pathways (gene sets). A statistical test is performed to ascertain whether the number of DEGs belonging to a particular gene set is higher than that expected by random chance, as determined by comparison to a background gene list. These ORA tools can be stand-alone software packages or web services, and they use one or more statistical tests (eg: Fisher's exact test, chi-square test) [3,4].

In the case of ORA for differential expression (eg: RNA-seq), a whole genome background is inappropriate because in any tissue, most genes are not expressed and therefore have no chance of being classified as DEGs. A good rule of thumb is to use a background gene list consisting of genes detected in the assay at a level where they have a chance of being classified as DEG [5–7]. Using the whole genome background gene list may be suitable in cases where all genes have the capacity of being detected, for example in studies of genetic variation (eg: [8]). However the problem becomes more acute when the proportion of measured genes/proteins is small, for example in proteomics and single-cell RNA-sequencing where only a few thousand analytes are detected.

FCS tools involve giving each detected gene a differential expression score and then evaluating whether the scores are more positive or negative than expected by chance for each gene set. The popular Gene Set Enrichment Analysis (GSEA) tool uses permutation approaches to establish whether a gene set is significantly associated with higher or lower scores, either by permuting sample labels or by permuting genes in the differential expression profile [9].

From a user's perspective, ORA is easier to conduct because it is as simple as pasting a list of gene names into a text box on a website. FCS tools are more difficult to use but are reported to have superior sensitivity in detecting subtle associations [10–13].

Although these are powerful tools to summarize complex genomics data, there are limitations. For example many ORA and FCS approaches assume independence between genes, which is problematic, as genes of the same functional category are somewhat more likely to have correlated gene expression [14]. There is an ongoing debate as to whether ignoring non-independence is a reasonable simplifying assumption in functional enrichment analysis [15,16]. Another issue is the reporting of statistically significant enrichments, where the observed effect size (enrichment score) is so small it is unlikely to have any meaningful biological effect [17].

Furthermore, there are concerns that enrichment tools are not being correctly used. Previous publications have warned that inappropriate background set selection heavily influences enrichment results [5,6]. Timmons et al [18] highlight two cases where an inappropriate background list led to invalid enrichment results in published articles.

While the nominal p-value from the gene set test is appropriate when a single set is examined, functional enrichment analysis typically involves hundreds to thousands of parallel tests, one for each gene set in the library (eg: MSigDB v7.4 library contains 32,284 sets [19]). False discovery rate (FDR) correction of enrichment p-values is therefore required to limit the number of false positives when performing so many concurrent tests [5–7,20].

Lack of methodological detail can severely weaken reproducibility. In 2001, minimum information about a microarray experiment (MIAME) guidelines were described [21] and rapidly adopted, resulting in higher reporting standards in publications describing microarray data. Reporting standards for computational biology have been proposed [22,23], but are not widely adopted. At a minimum, functional enrichment analysis reports should describe the methods used in such detail that the analysis could be replicated.

The main purpose of this work is to survey the frequency of methodological and reporting flaws in the literature, in particular; (i) inappropriate background gene set, (ii) lack of p-value adjustment for multiple comparisons and (iii) lack of essential methodological details. Secondly, we examine several RNA-seq datasets to evaluate the effect of such issues on the functional enrichment results obtained.

## Results

### Methodological and reporting deficiencies are widespread in articles describing functional enrichment analysis

A search of PubMed Central showed 2,941 open-access articles published in 2019 with the keywords "enrichment analysis", "pathway analysis" or "ontology analysis". From these, we randomly selected 200 articles for detailed methodological analysis. We excluded 14 articles from the screen because they did not present any enrichment analysis. Those excluded articles included articles describing novel enrichment analysis techniques or tools, review articles or conference abstracts. As some articles included more than one enrichment analysis, the dataset included 235 analyses from 186 articles; this data is available in **S1 Table**. A flow diagram of the survey is provided in **Fig 1**.

There were articles from 96 journals in the sample, with *PLoS One*, *Scientific Reports*, and *PeerJ* being the biggest contributors (**S1 Fig**). There were 18 different omics types, with gene expression array and RNA-seq being the most popular (**S1 Fig**). There were 31 different species under study, but *Homo sapiens* was the most common with 157 analyses (**S1 Fig**).

We recorded the use of 26 different gene set libraries, with GO and KEGG being the most frequently used (**Fig 2A**). There were 14 analyses where the gene set libraries used were not defined in the article. Only 18 analyses reported the version of the gene set library used (**Fig 2B**). There were 12 different statistical tests used, and the most commonly reported tests were Fisher's exact, GSEA and hypergeometric tests; but the statistical test used was not reported for most analyses (**Fig 2C**). Fourteen analyses did not conduct any statistical test, as they only showed the number of genes belonging to different sets. Out of the 221 analyses that performed a statistical test, only 119 (54%) described correcting p-values for multiple testing (**Fig 2D**).

There were 50 different tools used to perform enrichment analysis, with DAVID and GSEA being the most common; while 15 analyses (6.4%) did not state what tool was used (**Fig 2E**). The version of the software used was provided in only 68 of 235 analyses (29%) (**Fig 2F**).

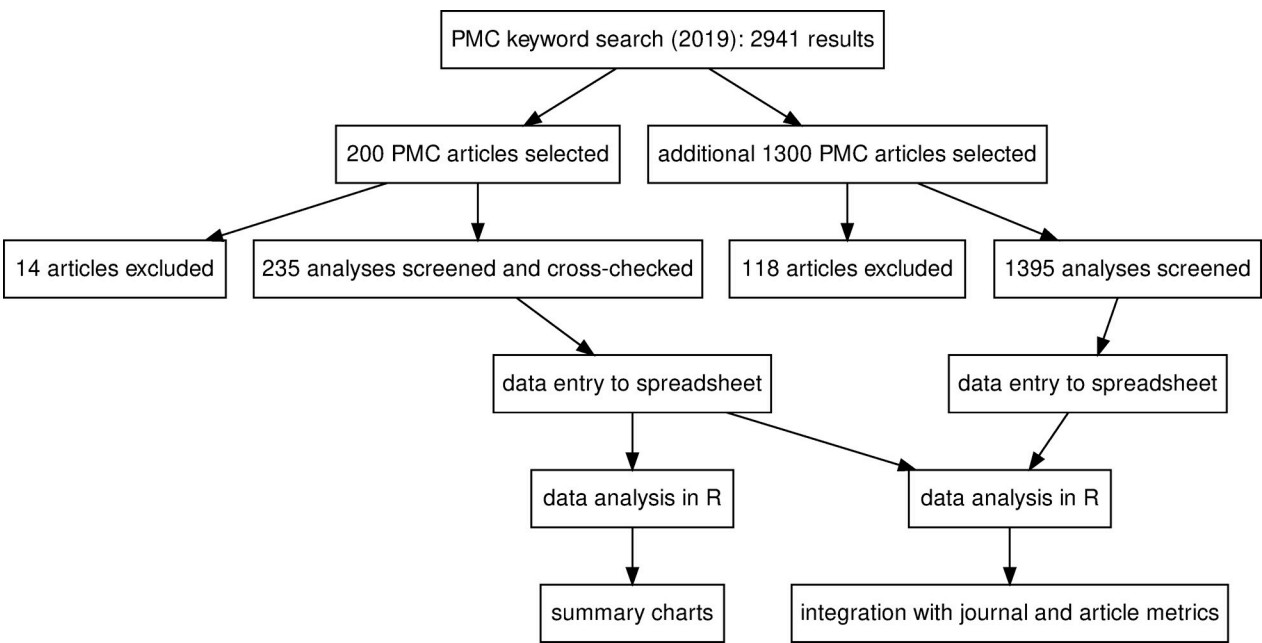

**Fig 1. A summary of the survey of functional enrichment analyses.** The survey consists of two parts, with 200 and 1300 PMC articles considered, respectively.

For analyses using ORA methods, we examined what background gene set was used (**Fig 2G**). This revealed that in most cases, the background list was not defined, or it was clear from the article methods section that no specific background list was used. In a few cases, a background list was mentioned but was inappropriate, for example using a whole genome background for an assay like RNA-seq. In only 8/197 of cases (4.1%), the appropriate background list was described in the article.

Of the 47 analyses which used computer scripts, only 3 provided links to the code used for enrichment analysis (6.4%) (**Fig 2H**). For 93 of 235 analyses (40%), the corresponding gene lists/profiles were provided either in the supplement or in the article itself (**Fig 2I**).

Next, we quantified the frequency of methodological and reporting issues that would undermine the conclusions (**Fig 2J**). Lack of appropriate background was the most common issue (179 cases), followed by lack of FDR control (94), then lack of data shown (13), inference without test (11), and misinterpreted FDR values (2). Only 35 analyses (15%) did not exhibit any of these major methodological issues.

We also looked at studies performing GSEA, and whether three important analytical choices were described in the methods. These are (i) the gene weighting parameter, (ii) test type, ie: permuting sample labels or genes, and (iii) method used for ranking genes. These parameters were not stated in more than half of the analyses (**Fig 2K**).

Taken together, these data suggest methodological and reporting deficiencies are widespread in published functional enrichment analyses.

## Methodological and reporting deficiencies occur irrespective of journal rank or article citations

To make an accurate assessment of association between analysis quality and bibliometrics, we required a larger sample of articles. We therefore screened a further 1,300 articles, bringing the total number of analyses described up to 1630; this dataset is available in **S2 Table**.

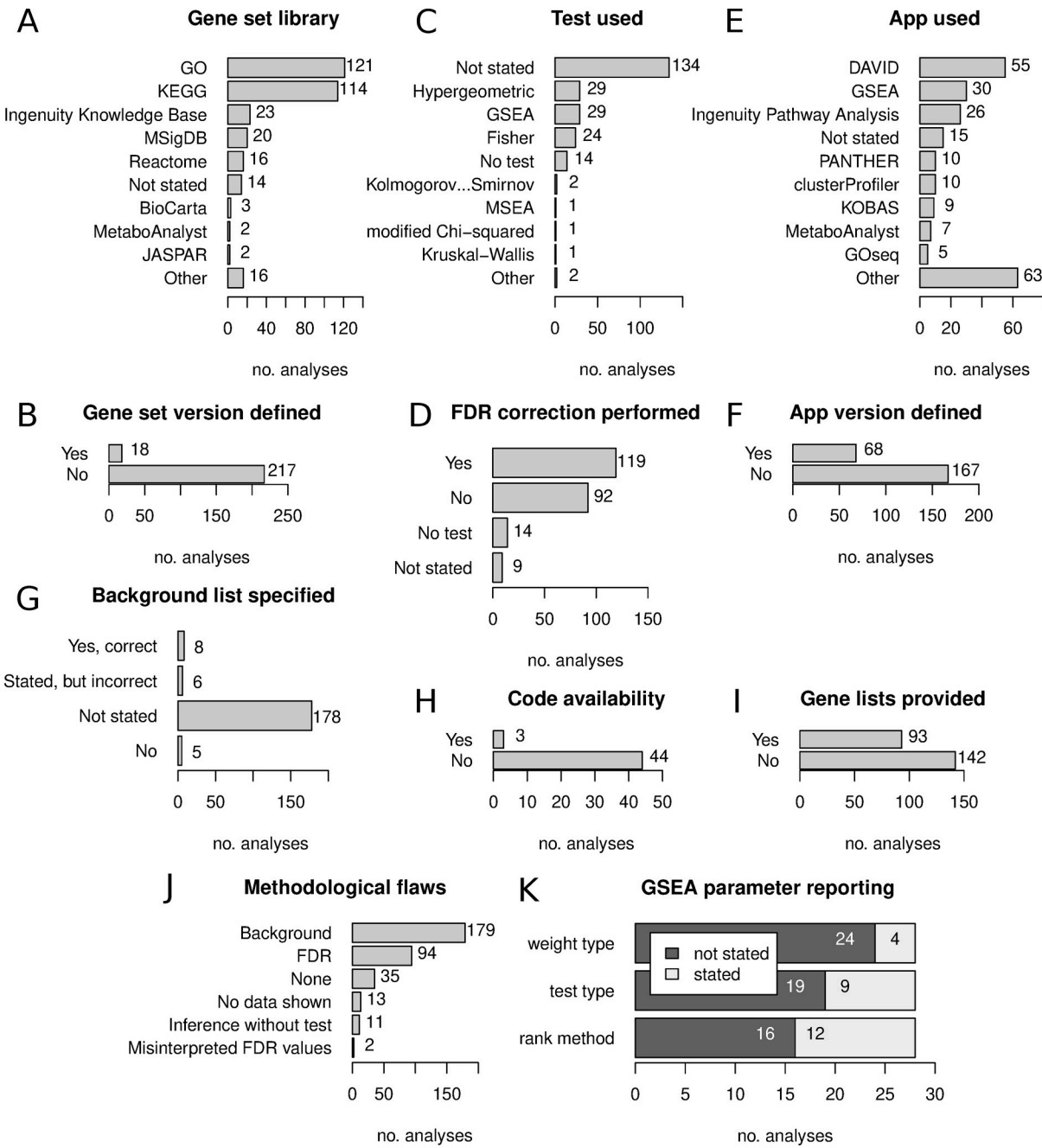

**Fig 2. Findings from the survey of published enrichment analyses.** (A) Representation of different gene set libraries. (B) Proportion of analyses reporting gene set library version information. (C) Representation of different statistical tests. (D) Proportion of analyses conducting correction of p-values for multiple comparisons. (E) Representation of different software tools for enrichment analysis. (F) Proportion of analyses that reported version information for software used. (G) Background gene set usage and reporting. (H) Proportion of scripted analyses that provided software code. (I) Proportion of analyses that provided gene profiles. (J) Proportion of analyses with different methodological problems. (K) Reporting of GSEA parameters.

We then scored each analysis based on the presence or absence of methodological issues and included details. The median score was -4, with a mean of -3.5 and standard deviation of 1.4 (**Fig 3A**). Next, we assessed whether these analysis scores were associated with Scimago Journal Rank (SJR), a journal-level citation metric. There was a slight positive association between analysis score

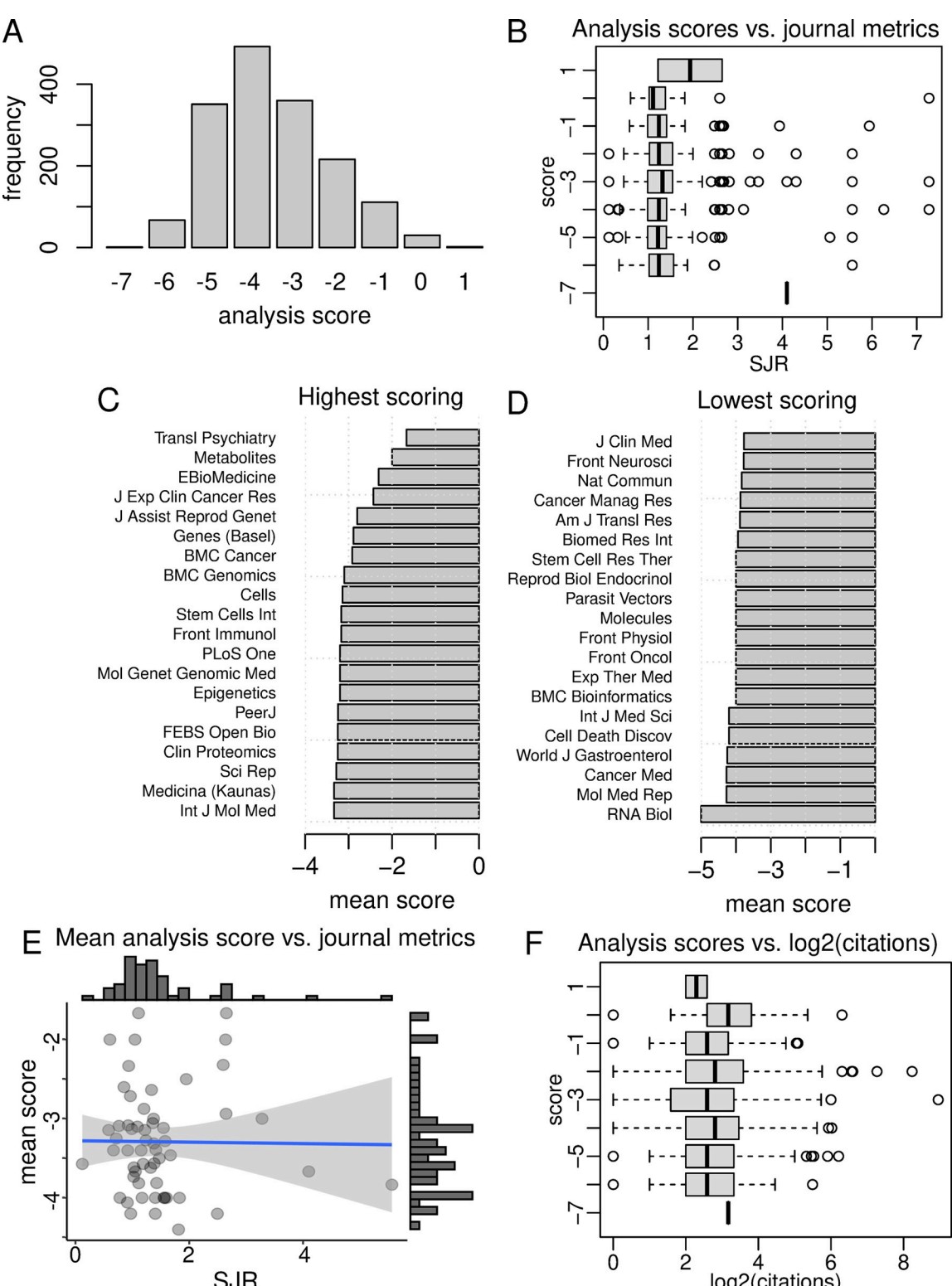

**Fig 3. Comparison of analysis scores to bibliometrics.** (A) Distribution of analysis scores. (B) Association of analysis scores with Scimago Journal Rank (SJR). (C) Journals with highest mean analysis scores. (D) Journals with lowest mean analysis scores. (E) Association of mean analysis score with SJR for journals with 5 or more analyses. (F) Association of analysis scores with accrued citations.

and SJR (Pearson r = 0.058, p = 0.036), but when viewed as boxplots of analysis score categories, there was no clear association between methodological rigor and SJR (**Fig 3B**).

Next, we wanted to know which journals had the highest and lowest scores. Only journals with five or more analyses were included. The best scoring journals were *Transl Psychiatry*, *Metabolites* and *J Exp Clin Cancer Res* (**Fig 3C**), while the poorest were *RNA Biol*, *Mol Med Rep*, *Cancer Med* and *World J Gastroenterol* (**Fig 3D**), although we note that there was a wide variation between articles of the same journal.

Then we assessed for an association between mean analysis score and the SJR, for journals with five or more analyses (**Fig 3E**). Again, there was no association between mean analysis score and SJR (Pearson r = -0.012, p = 0.93). Next, we assessed whether there was any association between analysis scores and the number of citations received by the articles. After log transforming the citation data, there was no association between citations and analysis scores (Pearson r = 0.02, p = 0.39) (**Fig 3F**). These findings suggest that methodological issues are not limited to lower ranking journals or poorly cited articles.

## Misuse of functional enrichment tools changes results substantially

To demonstrate whether functional enrichment analysis misuse affects results and downstream interpretation, we used an example RNA-seq dataset examining the effect of high glucose exposure on hepatocytes (SRA accession SRP128998). Out of 39,297 genes in the annotation set, 15,635 were above the detection threshold ($\geq$10 reads per sample on average). Statistical analysis revealed 3,472 differentially expressed genes with 1,560 up-regulated and 1,912 down-regulated (FDR<0.05) due to high glucose exposure.

To quantify the effect of not performing correction for multiple testing on enrichment results, FCS and ORA were performed and filtered at the nominal p<0.05 and FDR<0.05 levels. The number of gene sets found with each approach is shown in **Fig 4A**. This revealed that if p-value adjustment is ignored, then 25% of FCS and 39% of ORA results would erroneously appear as significant.

The overlap of significant gene sets (FDR<0.05) identified with FCS and ORA methods is shown in **Fig 4B**, and is reflected by a Jaccard statistic of 0.65, indicating moderate concordance between these methods.

We then performed ORA using a background list consisting of all genes in the annotation set, not just those detected by the assay (indicated as ORA*), which resulted in 139 up and 484 down regulated gene sets (FDR<0.05). The overlap of ORA with ORA* was relatively small, with a Jaccard statistic of 0.44 (**Fig 4C**). We then conducted ORA* without p-value adjustment (indicated as ORA*nom). As expected, the overlap between ORA and ORA*nom was very low, at 0.38 (**Fig 4D**).

This analysis was repeated with an additional six independent RNA-seq datasets for confirmation (**Fig 4E** and **Table 1**). The Jaccard values obtained confirm a consistent degree of similarity between FCS and ORA (mean Jaccard = 0.58). Across all studies, the lack of p-value adjustment and incorrect background affected results. Interestingly the effect of ignoring p-value adjustment was more acute for ORA as compared to FCS. Also, when it comes to ORA, the impact of incorrect background was observed to be more severe than ignoring p-value adjustment. As expected, results were drastically impacted when both methodological errors were present. In that case, the Jaccard index was <0.38 when compared to ORA with the recommended procedure.

## Discussion

Concerns have been raised that some articles describing enrichment analysis suffer from methodological problems [18], but this is the first systematic examination of the frequency of such issues in peer-reviewed publications.

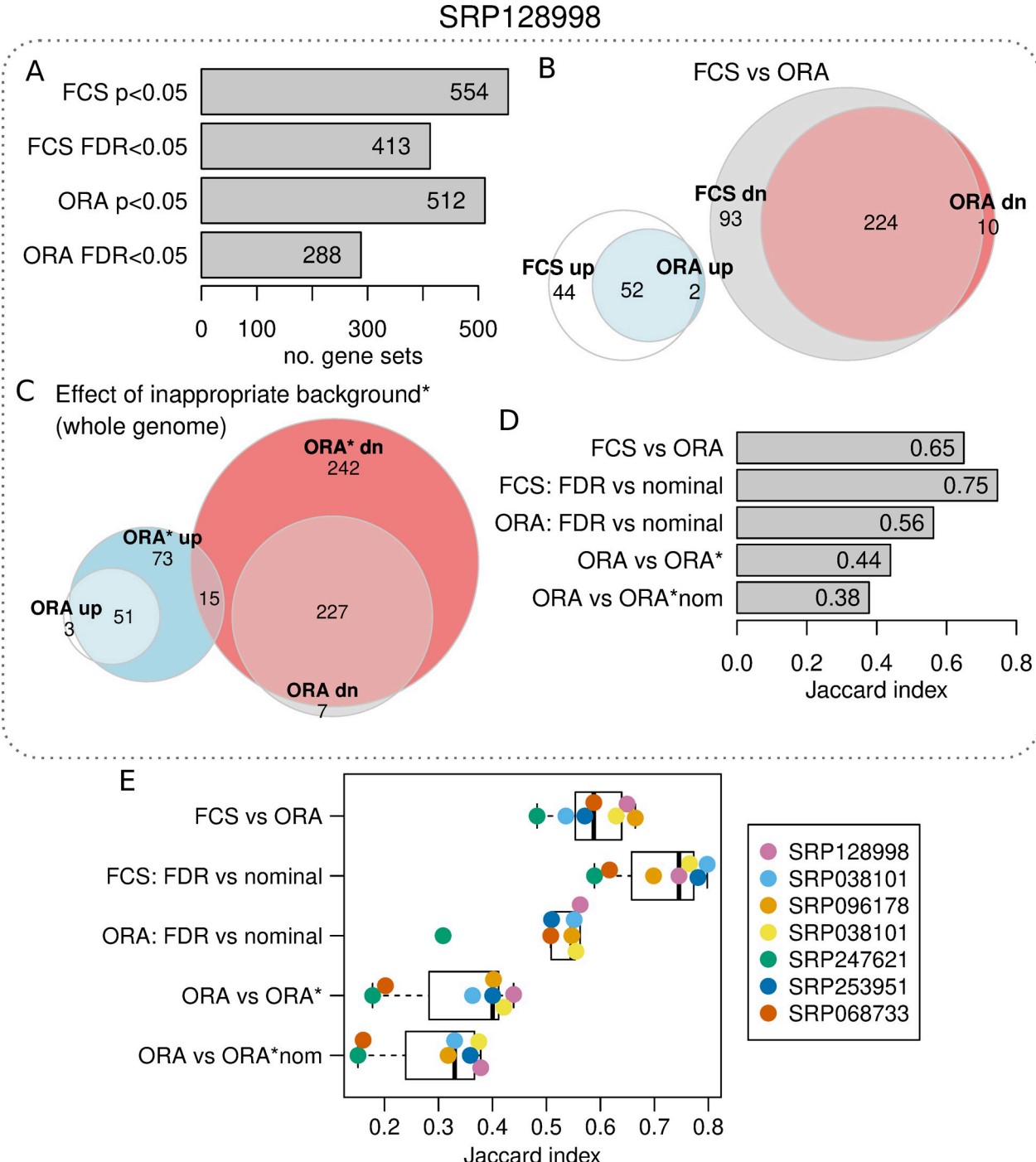

**Fig 4. Example enrichment analysis misuse.** (A) Number of differentially expressed gene sets for FCS and ORA methods using nominal p-value and FDR thresholds. (B) Euler diagram showing the overlap of differentially regulated gene sets with FCS and ORA methods (FDR<0.05). (C) Euler diagram shows the overlap of ORA results when using recommended or whole genome background. Whole genome background analysis is indicated with a *. (D) Jaccard index values for enrichment analysis results when conducted in different ways. "FDR" refers to a significance threshold of FDR<0.05. "Nominal" refers to a significance threshold of p<0.05. "ORA*nom" refers to a whole genome background used in tandem with the p<0.05 significance threshold. (E) Jaccard index values for enrichment analysis results when conducted in different ways for seven independent RNA-seq experiments.

**Table 1. Seven independent RNA-seq experiments used for functional enrichment analysis.** Detection threshold is an average of 10 reads per sample. Differentially expressed genes are defined as FDR<0.05 using DESeq2.

| SRA accession and citation | Control datasets | Case datasets | Genes detected | Genes differentially expressed |
|---|---|---|---|---|
| SRP128998 [24] | GSM2932797 GSM2932798 GSM2932799 | GSM2932791 GSM2932792 GSM2932793 | 15635 | 3472 |
| SRP038101 [25] | GSM1329862 GSM1329863 GSM1329864 | GSM1329859 GSM1329860 GSM1329861 | 13926 | 3589 |
| SRP037718 [26] | GSM1326472 GSM1326473 GSM1326474 | GSM1326469 GSM1326470 GSM1326471 | 15477 | 9488 |
| SRP096177 [27] | GSM2448985 GSM2448986 GSM2448987 | GSM2448982 GSM2448983 GSM2448984 | 15607 | 5150 |
| SRP247621 [28] | GSM4300737 GSM4300738 GSM4300739 | GSM4300731 GSM4300732 GSM4300733 | 14288 | 230 |
| SRP253951 [29] | GSM4462339 GSM4462340 GSM4462341 | GSM4462336 GSM4462337 GSM4462338 | 15182 | 8588 |
| SRP068733 [30] | GSM2044431 GSM2044432 GSM2044433 | GSM2044428 GSM2044429 GSM2044430 | 14255 | 7365 |

In this sample of open-access research articles, we observed a bias toward tools that are easy to use. ORA tools that only require pasting lists of gene identifiers into a webpage (ie: DAVID, KOBAS and PANTHER) were collectively more popular than other solutions like GSEA (a stand-alone graphical user interface software for FCS) or any command line tool (consistent with another report [13]). This is despite ORA tools being reported to lack sensitivity to detect subtle association according to previous benchmarking studies [10–13].

Failing to properly describe the background gene list was the most common methodological issue (**Fig 2J**). In the seven RNA-seq examples examined here, using the inappropriate whole genome background gave results that were on average only 44% similar to results obtained using the correct background (**Fig 4E**).

The severe impact of selecting the wrong background on RNA-seq functional enrichment results is due to the fact that in RNA-seq, typically only a small fraction of all annotated genes are detected. **Table 1** indicates that only ~38% of genes are detected in the seven examples. In contrast, a modern microarray detects a larger proportion of genes [31], so the effect of using a whole genome background is less severe.

There are various approaches to define the background of an RNA-seq dataset [32]. The effect of different filtering methods on differential expression results has been investigated [33], and provides us with some practical recommendations to avoid sampling biases described by Timmons et al [18].

Although articles that used GSEA obtained higher methodology scores overall, they were not free of issues. For example, GSEA has different options that impact the results including the ranking metric, gene weighting method and the permutation type (on samples or genes), which were not regularly reported in articles (**Fig 2K**), limiting reproducibility.

We scored a total of 1630 analyses, revealing only a small fraction that obtained a satisfactory score of zero or higher. The analysis scores we generated did not correlate with journal or article metrics. This suggests that methodological and reporting problems are not limited to lower ranked journals but are a more general problem.

These shortcomings are understandable, as some popular web tools do not accommodate background gene lists by design (eg: [34]). Moreover some user guides gloss over the problems of background list and correction for multiple testing (eg: [35]), while other guides are written in such a way that they are difficult for the novice statistician to comprehend (eg: [36]). Certainly the inconsistent nomenclature used in different articles and guides makes it difficult for beginners to grasp these concepts. Unfortunately, some of the best guides for enrichment analysis are paywalled (eg: [5,7]) which limits their accessibility. With this in mind, there is a need for a set of minimum standards for enrichment analysis that is open access and written for the target audience (life scientists with little expertise in statistics).

There are some limitations of this study that need to be recognized. Many open-access articles examined here are from lower-ranked journals that might not be representative of articles in paywalled journals. The articles included in this study contained keywords related to functional enrichment in the abstract, and it is plausible that articles in higher ranked journals contain such details in the abstract at lower rates. Those highly ranked specialist genomics journals are likely to have lower rates of problematic articles due to more knowledgeable editors and peer reviewers.

We also recognize the simplistic nature of the analysis scoring criteria. Clearly, the impact of each criterion is not the same. For example the effect of ignoring FDR is likely more severe than omitting the version number of the tool used. This simplified scheme was used for practical reasons.

Further, it is difficult to ascertain whether these methodological issues invalidate the conclusions of these articles. We are currently working on a systematic large scale replication study to determine the reliability of these articles with corrected methods.

In conclusion, these results are a wake-up call for reproducibility and highlight the urgent need for minimum standards for functional enrichment analysis.

## Methods

### Survey of published enrichment analysis

We collated 2,941 articles in PubMed Central published in 2019 that have keywords "enrichment analysis", "pathway analysis" or "ontology analysis". We initially sampled 200 of these articles randomly using the Unix "shuf" command. We then collected the following information from the article, searching the methods sections and other parts of the article including the supplement.

- Journal name

- Type of omics data

- Gene set library used, and whether a version was reported

- Statistical test used

- Whether p-values were corrected for multiple comparisons

- Software package used, and whether a version was reported

- Whether an appropriate background gene set was used

- Code availability

- Whether gene profile was provided in the supplement

- Whether the analysis had any major flaws that might invalidate the results. This includes:

  i. background gene set not stated or inappropriate background set used,

  ii. lack of FDR correction,

  iii. no enrichment data shown,

  iv. inference without performing any statistical test, and

  v. misinterpreting p-values by stating results were significant when FDR values indicate they weren't.

**Table 2. Scoring schema.**

| 1 point deducted | 1 point awarded |
|---|---|
| Gene set library origin not stated | Code made available |
| Gene set library version not stated | Gene profile data provided |
| Statistical test not stated | |
| No statistical test conducted | |
| No FDR correction conducted | |
| App used not stated | |
| App version not stated | |
| Background list not defined | |
| Inappropriate background list used | |

We excluded articles describing novel enrichment analysis techniques/tools, review articles and conference abstracts. Some articles presented the results of >1 enrichment analysis, so additional rows were added to the table to accommodate them. These data were entered into a Google Spreadsheet by a team of five researchers. These articles were cross checked by another team member and any discrepancies were resolved.

For analyses using GSEA, we scanned the articles to identify whether key methodological steps were described, including (i) the gene weighting parameter, (ii) test type, ie: permuting sample labels or genes, and (iii) method used for ranking genes.

For assessment of enrichment analysis quality with journal metrics and citations, we required a larger sample, so we selected a further 1300 articles from PMC for analysis. Results from this sample were not double-checked, so may contain a small number of inaccuracies. We rated each analysis with a simple approach that deducted points for methodological problems and missing details, while awarding points for including extra information (**Table 2**).

SJR data for 2020 were downloaded from the Scimago website (accessed 5th August 2021) and used to score journals by their citation metrics. Using NCBI's Eutils API, we collected the number of citations each article accrued since publication (accessed 3rd December 2021). Citation data were $\log_2$ transformed prior to regression analysis. Pearson correlation tests were used to assess the association with the analysis scores we generated.

## Exemplifying functional enrichment analysis misuse

To demonstrate the effect of misusing functional enrichment analysis, a publicly available RNA-seq dataset (SRA accession SRP128998) was downloaded from DEE2 on 19th January 2022 [37]. This data consists of immortalized human hepatocytes cultured in standard (n = 3) or high glucose media (n = 3), first described by Felisbino et al [24]. Transcript level counts were aggregated to genes using the getDEE2 R package v1.2.0. Next, genes with an average of less than 10 reads per sample were omitted from downstream analysis. Differential expression statistical analysis was conducted with DESeq2 v1.32.0 [38] to identify genes altered by high glucose exposure. For gene set analysis, human Reactome gene sets [39] were downloaded in GMT format from the Reactome website (accessed 7th December 2021). FCS was performed using the mitch R package v1.4.1 with default settings, which uses a rank-ANOVA statistical test [10]. Differentially expressed genes with FDR<0.05 were used for ORA analysis using the clusterProfiler R package (v4.0.5) enricher function that implements a hypergeometric test [40]. No fold-change threshold was used to select genes for ORA. For ORA, two types of background gene sets were used: (i) detected genes, or (ii) all genes in the genome annotation set. For genes and gene sets, a false discovery rate adjusted p-value (FDR) of 0.05 was considered

significant. Analyses were conducted in R version 4.1.2. To understand whether these results are consistent across other experiments, we repeated this analysis for an additional six independent published RNA-seq studies [25–30]. Details of the contrasts examined are shown in **Table 1**.

## Supporting information

**S1 Table. A survey of 186 articles describing functional enrichment results in TSV format.**
(TSV)

**S2 Table. A survey of 1300 articles describing functional enrichment results in TSV format.**
(TSV)

**S1 Fig. General information about the analyses that underwent screening.** (A) The most highly represented journals in the article set. (B) The most highly represented omics types used for enrichment analysis. (C) The most highly represented organisms under study.
(EPS)

## Acknowledgments

We thank Drs Antony Kaspi (Walter and Eliza Hall Institute), Nick Wong and Anup Shah (Monash University) for comments on the manuscript. This research was supported by use of the Nectar Research Cloud, a collaborative Australian research platform supported by the NCRIS-funded Australian Research Data Commons (ARDC).

## Author Contributions

**Conceptualization:** Mark Ziemann.

**Data curation:** Kaumadi Wijesooriya, Sameer A. Jadaan, Kaushalya L. Perera, Tanuveer Kaur, Mark Ziemann.

**Formal analysis:** Kaumadi Wijesooriya, Mark Ziemann.

**Investigation:** Kaumadi Wijesooriya, Sameer A. Jadaan, Kaushalya L. Perera, Tanuveer Kaur, Mark Ziemann.

**Methodology:** Kaumadi Wijesooriya, Mark Ziemann.

**Project administration:** Mark Ziemann.

**Resources:** Mark Ziemann.

**Software:** Kaumadi Wijesooriya, Mark Ziemann.

**Supervision:** Mark Ziemann.

**Validation:** Kaumadi Wijesooriya, Sameer A. Jadaan, Kaushalya L. Perera, Tanuveer Kaur, Mark Ziemann.

**Visualization:** Kaumadi Wijesooriya, Mark Ziemann.

**Writing – original draft:** Kaumadi Wijesooriya, Mark Ziemann.

**Writing – review & editing:** Kaumadi Wijesooriya, Sameer A. Jadaan, Kaushalya L. Perera, Tanuveer Kaur, Mark Ziemann.

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
