## [Decision Letter · Decision Letter 0]

6 Jan 2022

Dear Dr Ziemann,

Thank you very much for submitting your manuscript "Urgent need for consistent standards in functional enrichment analysis" for consideration at PLOS Computational Biology. As with all papers reviewed by the journal, your manuscript was reviewed by members of the editorial board and by several independent reviewers. The reviewers appreciated the attention to an important topic. Based on the reviews, we are likely to accept this manuscript for publication, providing that you modify the manuscript according to the review recommendations.

Sincerely,

Melissa L. Kemp, Ph.D.

Associate Editor

PLOS Computational Biology

Ilya Ioshikhes

Deputy Editor

PLOS Computational Biology

[LINK]

Reviewer's Responses to Questions

**Comments to the Authors:**

Reviewer #1: Enrichment analysis is widely used to interpret high-throughput omics data in terms of functional categories and pathways. In this study the authors laboriously surveyed 1630 genomics papers to assess whether enrichment analyses are conducted properly. Of these papers, 186 were studied in detail as they were cross checked by at least two team members. Their main finding is that only 15% of screened analyses are conducted and documented properly. Main issues in the rest are not using or reporting background genes (95%), failure to correct for multiple testing (43%), pooling up and down regulated genes. The author also analyzed an RNA-seq data set to demonstrate that how enrichment results differ when analyses is done inappropriately. As the first large-scale survey of its kind, this article sounds the alarm, once again, on the widespread abuse of statistics by the biomedical and genomics research community. The incompetency and the carelessness call for urgent action to improve reproducibility.

Fig.3b and 3f probably could be more effectively using boxplots/violin plots by treating analysis score as categories. Overlapping dots, even with color gradient, are hard to interpret.

Annotation databases for some online tools are not updated frequently. For example, the database for DAVID is last update in 2016. See https://david.ncifcrf.gov/content.jsp?file=release.html The author could discuss how obsolete annotation affect the results of enrichment analysis.

Reviewer #2: Wijesooriya et al. Urgent need for consistent standards in functional enrichment analysis

General comments

The Ziemann team address a fundamental scientific issue in the field of bioinformatics. Building on earlier commentaries and case examples they systematically address and quantify the extent of the ‘erroneous pathway p-value issue’ in biomedical research. Their conclusions are not surprising to this reviewer, yet these results make urgent and essential reading for all reviewers, journal editors and biomedical researchers working with OMIC data.

There is one main and easy to solve flaw in their article and that pertains to an argument about combining or otherwise up and down-regulated genes. There is no logic or rule for this and in biological pathways combining is completely acceptable, as you can have both positive and negative regulators in a list from one pathway, differentially expressed in the opposite direction. Thus the interpretation of the statistical work by Hong G et al 2014 is wrong – unsurprisingly as they understand little biology based on a reading of their article.

Thus an up-regulation of a positive actor or down-regulated of a negative actor – in the same pathway - can equate to the same biochemical outcome. Splitting lists also actually leads to two other problem. First is that with some technologies detecting down-regulation is more difficult (signal related) and second is that related to gene list size. Its well appreciated that small lists, especially if enrichment ratios + pvalues, and/or boot strapping are considered, are not sensitively profiled. Splitting a biologically relevant list into up and down impacts on this issue in an unpredictable manner, depending on gene list size and content. This particular section needs re-written or deleted as its wrong. Your results – see below – actually stumble on this issue.

Specific comments

The majority of my comments are minor and relate to ensuring clarity of message and identification of any statement that could be misconstrued (or is not entirely accurate).

P3 ‘Instead, many tools have been developed to summarise gene profiles into simplified functional categories’

I would breifly mention the history behind the Gene Ontology Consortium and their project to provide context as to how this catalogue of processes/pathways emerged. http://geneontology.org/docs/introduction-to-go-resource/

Its possibly informative to reflect on the fact that this grew out of a need to catalogue the genome (with the incorrect assumption that all genes were equally characterizable in a genome wide study). So perhaps they never thought about a variable detectable background.

You should probably mention that the Gene Ontology Consortium current link to their “10 tips for GO” does not mention the word ‘background’ even once. https://journals.plos.org/ploscompbiol/article?id=10.1371/journal.pcbi.1003343

This is a paragraph from Chapter from the ‘The Gene Ontology HandBook’. 2017. They STILL don’t correctly explain the background bias issue.

“Such large changes in GO annotations can affect GO enrich- ment analyses, which are sensitive to the choice of background distribution (Chap. 13 [3]; [20]). For instance, Clarke et al. [21] have shown that changes in annotations contribute significantly to changes in overrepresented terms in GO analysis. To mitigate this problem, researchers should analyze their datasets using the most up-to-date version of the ontology and annotations, and ensure that the conclusions they draw hold across multiple recent releases. At the time of the writing of this chapter, DAVID, a popular GO analysis tool, had not been updated since 2009”

P3 ‘developed to summarise gene profiles into simplified functional categories’

Try – ‘developed to summarise regulated gene expression profiles into simplified functional categories’

P4 ‘A statistical test is performed to ascertain whether the number of DEGs belonging to a particular gene set is higher than that expected by random chance, as determined by comparison to a background gene list. These ORA tools can be stand-alone software packages or web services, and they use one or more statistical tests (eg: Fisher’s exact test, hypergeometric test) [1,2]’.

I have two comments here.

Most ignore the enrichment ratio (which helps inform about bias related to very large (usually uninformative) gene sets) e.g. significant p-value and ER of 1.1 is not a meaningful result (given all the sources of bias). I would explicitly mention that ‘is ideally both a significant adjusted p-value and a robust enrichment ratio (ratio of regulated genes to total genes in that group) is sought’.

Second point is that I believe it is essential to mention the published objections to use of Fisher’s exact test (or worse still hypergeometric test). In that they assume independence and for members of gene sets that not true. I would state there are “general concerns” about the weakness of the primary statistical methods. The reason I mention this is some colleagues won’t engage on the larger issue of background bias or lack of FDR use, because they object to the primary statistical method. Better to state you recognise their concerns with a sentence and citation.

A good review article to cite on the GO stats would be https://www.nature.com/articles/nrg2363

Also note this sentence from the review.

“In practice, a term would need to have a raw p-value less than 4 x 10−7 for it to be significant at the 1% significance level. Other corrections, such as Holm's26 and false discovery rate27, are less conservative but loss of power cannot be completely avoided (see Refs 28,29 for further reviews). Hence, as a general rule, one can increase the power of the statistical analysis by performing the fewest possible number of tests.”

This review deals with things are a theoretical level and do not do the essential work you provide in your article – namely the scale for the problem. It is worrying however how little of this review in 2008 has been recognised, compared with the use of GO tools.

Hammering home that a p-value of 1x10-6 BEFORE correction is likely to required will provide some reality check to those using GO tools and the designers of software – as they are the MAIN problem today. If your tool does not give good results people will not use them.....thats clearly an issue.

Secondly, many select EVERY GO class, or GO + KEGG + XYZ when using online tools - This is clearly a flawed approach, yet software tools enable this mistake.

A further point that you may wish to consider specifically in relation to microarrays and RNAseq. The modern microarray provides greater coverage (See Sood et al Nucleic Acid research 2016 and Timmons et al Aging Cell 2019 and supplement), and are more sensitive (See Fig 2e, Peters TJ Bioinformatics 2019) when profiling individual human tissues than RNAseq.

RNAseq has a library (PCR bias and incomplete nature means a gene can not ‘appear’) and also a serious reproducibility problem (See Supplement S22 of the Sequencing Quality Control Consortium, Nature Biotechnology 2014 – 32(9)) meaning that the certainty of the background becomes less clear than for a modern array processed with modern methods. See Mandelboum et al Plos Biology 2019).

Single Cell RNAseq becomes an even greater issue – here coverage is normally between 2000-5000 genes and heavily biased for high abundance genes esp. mitochondrial (See the Gene Ontology data buried in the supplement of Mereu et al Nature Biotechnology June 2020).

In short, for sequencing experiments the GO/Pathway background issue just became a far greater problem. This is important.

Page 6 “From these, we initially selected 200 articles for detailed methodological analysis.”

Please clarify why and how you selected the 200 for the summary chart arm of your flow chart.

I am curious that so few Nature Communication articles appear in your analysis. This is the largest single source of OMIC papers I encounter and almost without exception there is no clear methods and obvious flaws in the GO/Pathway analysis. The other journals that have a high frequency of flawed GO/Pathway analysis are the American Journal of Physiology family of journals.

In that sense Fig S1A is not helpful as its not an unbiased or full representation of the sources of the problem – and might give the wrong impression.

P8 During this survey, we noticed some studies grouped up- and down-regulated gene lists together prior to ORA, a practice we were not expecting.

As mentioned above you have made a mistake in logic, regarding up and down-regulated gene lists and their combination. See above.

P11. Figure 3D is remarkable – BMC Bioinformatics being one of the lowest scoring journals. I am glad I resigned from their editorial board 8yr ago (for poor quality editorial processes).

P12. The RNA-seq data you analyse using only an FDR filter and with >20% DE raises issues about normalisation if such a large % of genes are genuinely DE. I personally would check implementing a modest FC filter on top of the p-value to enrich in true positives and confirm that you still have only moderate concordance. In short, if you put a lot of ‘junk’ in, you can not expect reliable data out and its best to avoid any critic of this analysis.

P12 “Interestingly, 26 gene sets were simultaneously up and downregulated with this approach.”

This is probably a reflection of the issue of splitting up and down regulated lists as I mentioned. The direction does not come from GO, it comes from your assumption that up means “process up” and vice versa. This is not reliably concluded without detailed inspection as, as stated, loss of a negative regulator results in up-regulation of pathway function. You need to rethink this part of your paper otherwise you are jeopardising the validity of your article. Likewise you need to re think how you cite the flawed conclusions made by Hong 2014.

14. Hong G, Zhang W, Li H, Shen X, Guo Z. Separate enrichment analysis of pathways for up- and downregulated genes. J R Soc Interface. 2014;11: 20130950.

Page 15

“This is despite ORA tools being reported to lack sensitivity when compared to FCS according to previous benchmarking studies [7-9]”

“Although analyses involving GSEA scored better overall, they were not free of issues. “

You should probably consider that FCS/GSEA represent a different set of biases (depending on the origin and date of the gene-sets) than ORA GO type analysis. Arguing that FCS is more sensitive is partly a reflection of using the KS statistic and there are strong advocates against the robustness of KS (and several iterations trying to address them). On the other hand being able to show that the accumulation of a large number of small changes in a pathway might be biologically sound is attractive. I would present + and – rather than just that FCS is ‘better’ based on sensitivity.

Methods

List that the journal scoring metric is logical but ad hoc, and it is not scaled by the magnitude of impact of each error on the results.

Human nature is to say “I followed almost all the criteria so I am doing well” when they could miss out the most damaging rule e.g. No FDR and inappropriate background”. I’d state you are not implying that all scoring criteria are equal.

From peer reviewing the number of times I have informed an author that they get “relevant pathways” (to their tissue/biology) because they compare with a genome wide background and that creates fake enriched p-values – only to have this ignored and the fake enriched p-values published, is substantial.

Defining the correct RNAseq background is particularly challenging as many samples can have essentially zero counts for a particular gene (unrelated to group membership), and yet its called detected by some % call. Proportion calls with groups (block design) is required. You might wish to write something more about the sources background biases or cite from Timmons 2015, that lists them.

**Have the authors made all data and (if applicable) computational code underlying the findings in their manuscript fully available?**

Reviewer #1: Yes

Reviewer #2: Yes

PLOS authors have the option to publish the peer review history of their article (what does this mean?). If published, this will include your full peer review and any attached files.

Reviewer #1: **Yes: **Xijin Ge

Reviewer #2: No

Figure Files:

Data Requirements:

Reproducibility:

References:

---

## [Decision Letter · Decision Letter 1]

18 Feb 2022

Dear Dr Ziemann,

We are pleased to inform you that your manuscript 'Urgent need for consistent standards in functional enrichment analysis' has been provisionally accepted for publication in PLOS Computational Biology. The reviewers appreciated your attention to their critical comments and felt your revised, focused version would be an invaluable resource to the journal's readership.

Best regards,

Melissa L. Kemp, Ph.D.

Associate Editor

PLOS Computational Biology

Ilya Ioshikhes

Deputy Editor

PLOS Computational Biology

Reviewer's Responses to Questions

**Comments to the Authors:**

Reviewer #1: The authors have addressed my concerns. I also like it that the authors choose to focus on specific issues rather than attacking many things at the same time. The manuscript is much improved.

Regarding the pooling of up- and down-regulated genes for ORA, my opinion differs slightly from that of Reviewer #2. I think, in most cases, splitting up- and down-regulated genes are recommended. The two pitfalls (difficulty in detecting down-regulated genes, and list size) for splitting listed by reviewer #2 are not frequently encountered in RNA-Seq studies. Yes, positive and negative regulators of a pathway is possible sometimes. But pooling these gene lists introduces more noise and as far as I can see is not a common practice. I agree with a lot of excellent, constructive points of Reviewer #2. But authors should be allowed to express their opinions.

Reviewer #2: Dear Authors

Thanks for your responses and adjustments. I think the article is now in excellent shape.

One technical point, you are free to consider, is the fold-change filter. There is extensive studies from the older microarray field demonstrating the impact of a combined FC and P-value on FDR control for individual genes (See the work around the time of Choe et al). In terms of ontology analysis you can observe this effect empirically, yourself, if you have large list motivated by p-values only (e.g. n=1500), and you compare the GO profile obtained with and without a FC filter (e.g. down to n=750) you not a substantial impact on GO pathway enrichment (often losing large GO categories with modest enrichment ratios). In terms of FC filters, it is less arbitrary than p<0.05 - in that you decide to filter based on the technical performance of the platform. For an array that could be >10% shift in signal. For sequencing that is probably closer to 20%.

A FC filter would impact on your results but its up to you!

**Have the authors made all data and (if applicable) computational code underlying the findings in their manuscript fully available?**

Reviewer #1: Yes

Reviewer #2: Yes

PLOS authors have the option to publish the peer review history of their article (what does this mean?). If published, this will include your full peer review and any attached files.

Reviewer #1: No

Reviewer #2: No

---

## [Editor Report · Acceptance letter]

4 Mar 2022

PCOMPBIOL-D-21-02205R1 

Urgent need for consistent standards in functional enrichment analysis

Dear Dr Ziemann,

I am pleased to inform you that your manuscript has been formally accepted for publication in PLOS Computational Biology. Your manuscript is now with our production department and you will be notified of the publication date in due course.

With kind regards,

Anita Estes
